# An extreme mutational hotspot in *nlpD* depends on transcriptional induction of *rpoS*

**Andrew D. Farr**[1,2]*, **Christina Vasileiou**[1], **Peter A. Lind**[2,3,4], **Paul B. Rainey**[1,2,5]

**1** Department of Microbial Population Biology, Max Planck Institute for Evolutionary Biology, Plön, Germany, **2** New Zealand Institute for Advanced Study, Massey University, Auckland, New Zealand, **3** Department of Molecular Biology, Umeå University, Umeå, Sweden, **4** Umeå Centre for Microbial Research (UCMR), Umeå University, Umeå, Sweden, **5** Laboratoire Biophysique et Évolution, CBI, ESPCI Paris, Université PSL, CNRS, Paris, France

* afarr@evolbio.mpg.de

**Data availability statement:** Raw data and calculations performed to produced visualised data are available via Zenodo (https://doi.

## Abstract

Mutation rate varies within and between genomes. Within genomes, tracts of nucleotides, including short sequence repeats and palindromes, can cause localised elevation of mutation rate. Additional mechanisms remain poorly understood. Here we report an instance of extreme mutational bias in *Pseudomonas fluorescens* SBW25 associated with a single base-pair change in *nlpD*. These mutants frequently evolve in static microcosms, and have a cell-chaining (CC) phenotype. Analysis of 153 replicate populations revealed 137 independent instances of a C565T loss-of-function mutation at codon 189 (CAG to TAG (Q189*)). Fitness measures of alternative *nlpD* mutants did not explain the deterministic evolution of C565T mutants. Recognising that transcription can be mutagenic, and that codon 189 overlaps with a predicted promoter (*rpoSp*) for the adjacent stationary phase sigma factor, *rpoS*, transcription across this promoter region was measured. This confirmed *rpoSp* is induced in stationary phase and that C565T mutation caused significant elevation of transcription. The latter provided opportunity to determine the C565T mutation rate using a reporter-gene fused to *rpoSp*. Fluctuation assays estimate the C565T mutation rate to be ~5,000-fold higher than expected. In *Pseudomonas*, transcription of *rpoS* requires the positive activator PsrA, which we show also holds for SBW25. Fluctuation assays performed in a Δ*psrA* background showed a ~60-fold reduction in mutation rate confirming that the elevated rate of mutation at C565T mutation rate is dependent on induction of transcription. This hotspot suggests a generalisable phenomenon where the induction of transcription causes elevated mutation rates within defining regions of promoters.

## Authors summary

Genetic mutations drive evolution, with effects ranging from harmful to beneficial. While typically considered random, some genomic positions – called "mutational hotspots" – mutate more frequently. Here, we investigated a striking pattern of identical mutations repeatedly arising at the same position in replicated bacterial populations. The mutation lies within the promoter of *rpoS* (encoding the stationary phase sigma factor) and

org/10.5281/zenodo.14335473), with each folder of data corresponding to a different figure (including supporting information figures). Sequences and annotations of plasmids pUI-*sacB*, pTn7-*rpoSp-kan*, pTn7-gfp-*rpoSp-kan* and pTn7-scarlet-*rpoSp-kan* have been deposited in the European Nucleotide Archive (ENA) at EMBL-EBI under accession number PRJEB84419 (https://www.ebi.ac.uk/ena/browser/view/PRJEB84419).

**Funding:** ADF, CV and PBR acknowledge generous core funding from the Max Planck Society (https://www.mpg.de/en). The Max Planck Society had no role in the study design, data collection and analysis, decision to publish, or preparation of the manuscript.

**Competing interests:** The authors have declared that no competing interests exist.

overlaps with another gene, *nlpD*. Initially, we hypothesized that parallel evolution resulted from strong competitive growth driven by changes to either gene. However, competitive growth measurements did not explain the repeated evolution of these mutants. Instead, we found that parallel evolution was caused by an unusually high mutation rate, measured as orders of magnitude greater than expected. Further analysis revealed that increased promoter activity was linked to elevated mutation rates. These findings raise broader questions about whether similar hotspots exist in other bacterial species and how they might shape gene regulation and adaptation.

## Introduction

Mutation fuels evolution by natural selection. Despite the long-term benefits of mutation for life on earth, most mutations are neutral or deleterious [1], and the rate at which they occur is tightly regulated [2,3]. In bacteria, point mutations to any one base-pair occur approximately once every $1 \times 10^{10}$ cell divisions, with the lowest average mutation rates occurring in *Pseudomonas fluorescens* SBW25 [4]. However the rate of mutation varies significantly between regions of genomes (extensively reviewed [5]), with some nucleotide positions – referred to as "mutational hotspots" – subject to high frequency change.

The mechanistic causes of mutational hotspots are various and idiosyncratic. Simple sequence repeats can undergo slipped-strand mis-repairing [6], resulting in high-frequency gain or loss of repeats [7]. Imperfect inverted repeat sequences ('quasi-palindromes') may induce template-switching, with a range of mutational effects [8]. Other hotspots emerge from interactions with transcriptional machinery. In a study of mutation rates at *thyP3* in *Bacillus subtilis*, Sankar and colleagues measured high levels of thymine to cytosine point mutations (at a rate of ~$3 \times 10^{-8}$ per cell per generation) at a single base pair directly upstream of the transcription start site [9]. This was enhanced by activity of the promoter and orientation of the gene, leading to the suggestion that stalled RNA polymerase can allow deamination of nucleotides on template strands. Such specific mutagenic processes can interact with regional mutational biases, such as genomic strandedness and location, to enhance mutation rates [10,11], but also any process that biases the spectrum of mutations in the genome [12–14]. The multitude of mechanistic causes of hotspots has a singular consequence: they increase the production of variants presented to selection [15], often explaining instances of parallel evolution [16].

A remarkable example of parallel evolution was previously identified across multiple independent cultures and during independent studies using the model organism *Pseudomonas fluorescens* SBW25 [17,18]. Parallelism was evident from repeated observation of identical cytosine (C) to thymine (T) mutations at base pair 565 (C565T) of *nlpD*. The mutation alters codon 189 (sequence CAG – encoding glutamine) and results in a premature stop codon (sequence TAG). These mutants – arising in statically incubated microcosms of KB media due to a measurable fitness advantage over ancestral genotypes [17] – were identified by subtle changes to colony morphology and a cell-chaining (CC) phenotype. The CC phenotype is readily explained by the predicted function of NlpD – a protease that enables activation of amidases that cleave peptidoglycan during cellular division [19]. Depending on the presence of related genes, loss-of-function mutations in NlpD can result in incomplete cell separation in genera such as *Escherichia* [20] and *Pseudomonas* [19]. It was previously suggested that repeated detection of the C565T mutation in *nlpD* might be explained by a secondary function

encoded by *nlpD* [17], namely, that nested within *nlpD* is the primary promoter for the downstream gene, *rpoS* (we refer to the promoter of *rpoS* as '*rpoSp*') [21–23]. Assuming this mutation also alters transcription of *rpoS*, the combination of NlpD truncation and altered RpoS expression – resulting in increased fitness compared to both the ancestor and other *nlpD* mutants – might explain the repeated observation of C565T mutants in static microcosms [17].

In this study, we build upon previous work and demonstrate that a single nucleotide in *nlpD* is prone to mutate at high frequency. Assessment of the fitness of additional *nlpD* mutants shows that molecular parallelism at C565T is not caused by selection. Characterisation of the *rpoS* promoter led to construction of a mutational reporter to measure mutation rate at C565T with further work demonstrating that elevated mutation is transcription-dependent.

## Results

### Extreme molecular parallelism within *nlpD*

We previously reported mutational parallelism of the C565T mutation following adaptive radiation of the SBW25 Δ*wss* strain in statically incubated microcosms [17]. We wanted to understand the mechanistic basis of this molecular parallelism. However, we first repeated the original experiment but with greater replication: 153 static broth microcosms were inoculated with SBW25 Δ*wss* and screened for the presence of CC mutants. SBW25 Δ*wss* – devoid of the cellulose biosynthetic pathway – was used to limit the evolution of 'Wrinkly Spreader' morphotypes that overproduce cellulose that readily evolve in static microcosms inoculated with SBW25 [24] and otherwise obscure the detection of CC types [17]. The cultures were diluted, spread on agar plates, and subsequent colonies were screened for the aberrant cellular chaining (CC) morphology characteristic of *nlpD* mutants. From 153 populations, CC mutants were detected in 139 populations, with sequencing showing that 137 contained the C565T mutation (Fig 1A), one a G539A mutation (resulting in a G180D non-synonymous mutation) and one a frame shift (fs) insertion inside a homopolymeric tract (base pairs 705-708; Q237fs). While additional mutations to *nlpD* can arise in static microcosms, the predominant cause of the CC type was the C565T mutation, resulting in an extreme degree of molecular parallelism.

A previous study [18] had serendipitously obtained a *nlpD* mutant that firstly acquired the C565T mutation, followed by a second mutation at nucleotide 566 (A566G) delivering a CAG to TGG sequence change at codon 189 of *nlpD*. This codon change resulted in a glutamine to tryptophan substitution – Q189W – at codon 189. Given that the TGG mutation at codon 189 was not CC [18], we asked whether CC mutants might still arise by disproportionate changes at codon 189 caused specifically by reverting G566A mutations (resulting in a sequence change of TGG back to TAG). To this end, codon 189 of *nlpD* in SBW25 Δ*wss* was altered from CAG to TGG and the resulting mutant propagated in 54 static broth microcosms. CC mutants were detected in 38 microcosms, with 37 harbouring mutations in *nlpD*. Of these, eight acquired the reverting G566A mutation (resulting in TAG stops at codon 189), with the remaining mutations scattered throughout the gene (Fig 1B and S1 Table). This shows that a subtle change in the sequence of codon 189 dramatically reduces (but does not abolish) mutational parallelism across replicate microcosms.

### Fitness of C565T mutants does not sufficiently explain parallel mutation

Armed with a suite of additional CC mutants carrying defects in *nlpD* other than C565T, it was possible to ask whether the C565T mutation conferred a selective advantage that might explain its repeated occurrence. Considering both the near-deterministic occurrence of the

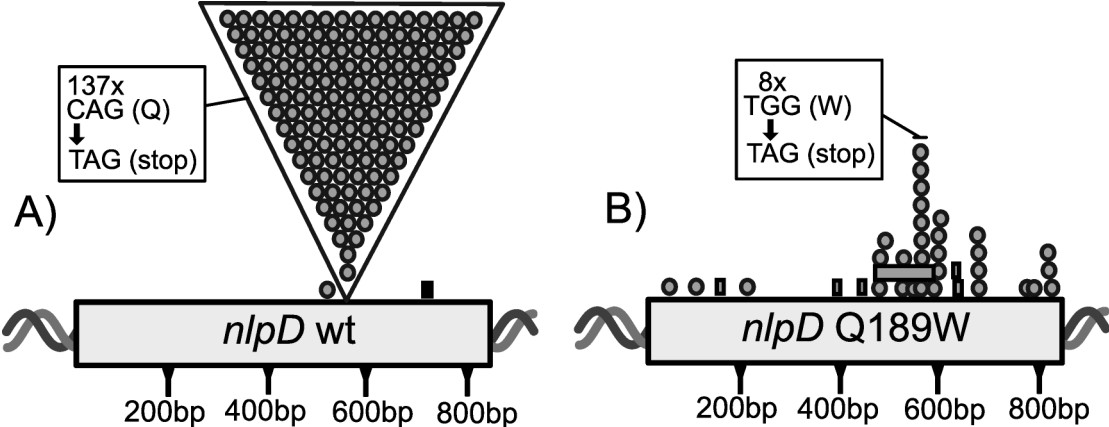

**Fig 1. The position of CC mutations found within *nlpD*.** A) Distribution of *nlpD* mutants evolved from SBW25 Δ*wss*. Depicted is the position of *nlpD* mutants evolved from SBW25 Δ*wss* cultured in static microcosms. The majority (137) of mutants were identical C to T mutations (gray circles in large triangle) at base pair 565 of *nlpD* (C565T). The C565T mutation results in a CAG to TAG nucleotide sequence change at codon 189 (Q189*; see box). Two alternative non-synonymous mutations were identified, a point mutation and an insertion (black rectangle). B) Distribution of *nlpD* mutants evolved from SBW25 Δ*wss nlpD* Q189W. This distribution of mutations changes when the founding genotype is SBW25 Δ*wss nlpD* C565T A566G (Q189W), with the nucleotide sequence of codon 189 being TGG. Evolution from this genotype resulted in point mutations (grey circles) and deletion mutations (grey rectangles) spread across the open reading frame of *nlpD*. A degree of molecular parallelism remains, with eight mutants having the G566A mutation (resulting in a TGG to TAG sequence change at codon 189; see box).

C565T mutant and the near-absence of alternative *nlpD* mutants evolving from SBW25 Δ*wss*, we expected a strong fitness advantage of the C565T mutant over all possible *nlpD* truncation mutants. The fitness of six early termination mutations derived from the Q189W (TGG) mutant (S1 Table), as well as the Q237fs mutant evolved from SBW25 Δ*wss*, were compared to the C565T mutant. For genotypes derived from Q189W, codon 189 was restored to the ancestral CAG sequence, to eliminate any phenotypic effects of the Q189W mutation on the fitness of the *nlpD* truncation mutants growing in static microcosms. Five of the most 3' *nlpD* mutations all had significantly lowered fitness compared to the control competitions of C565T against itself (Fig 2; p < 0.0001; Welch's one-way ANOVA, $F(8,25.652) = 58.649$, p < 0.0001, pairwise difference assessed with Tukey HSD). This suggested the C565T mutation may confer a selective benefit over many other loss-of-function mutations to *nlpD*. However, the two 5' *nlpD* mutations (stop and frameshift mutations at codons 17 and 132) were not significantly different from the control. Strikingly, the mutation to codon 17 (a C49T mutation to *nlpD*) was caused by the same C to T transition mutation as C565T, and yet was never observed in evolution experiments with SBW25 Δ*wss*. The probability of observing 137 instances of the C565T mutant and 0 instances of C49T mutants (assuming the same fitness and mutation rate) is $0.5^{137} = 5.7 \times 10^{-42}$. Instead of competitive advantage, an alternative explanation was sought for the repeated evolution of the C565T mutant.

## Transcription from the *rpoS* promoter (*rpoSp*) is elevated in stationary phase and is strengthened by the C565T mutation

An alternative explanation for the near-deterministic evolution of the C565T mutants is that the region around codon 189 of *nlpD* mutates at a high rate, resulting in the C565T mutation being presented to selection at a high frequency. Could the presence of *rpoSp* – inferred as proximal to the mutation from promoter mapping in other pseudomonads [21,22] – define a mutational hotspot? Confirmation and characterisation of this promoter could also prove

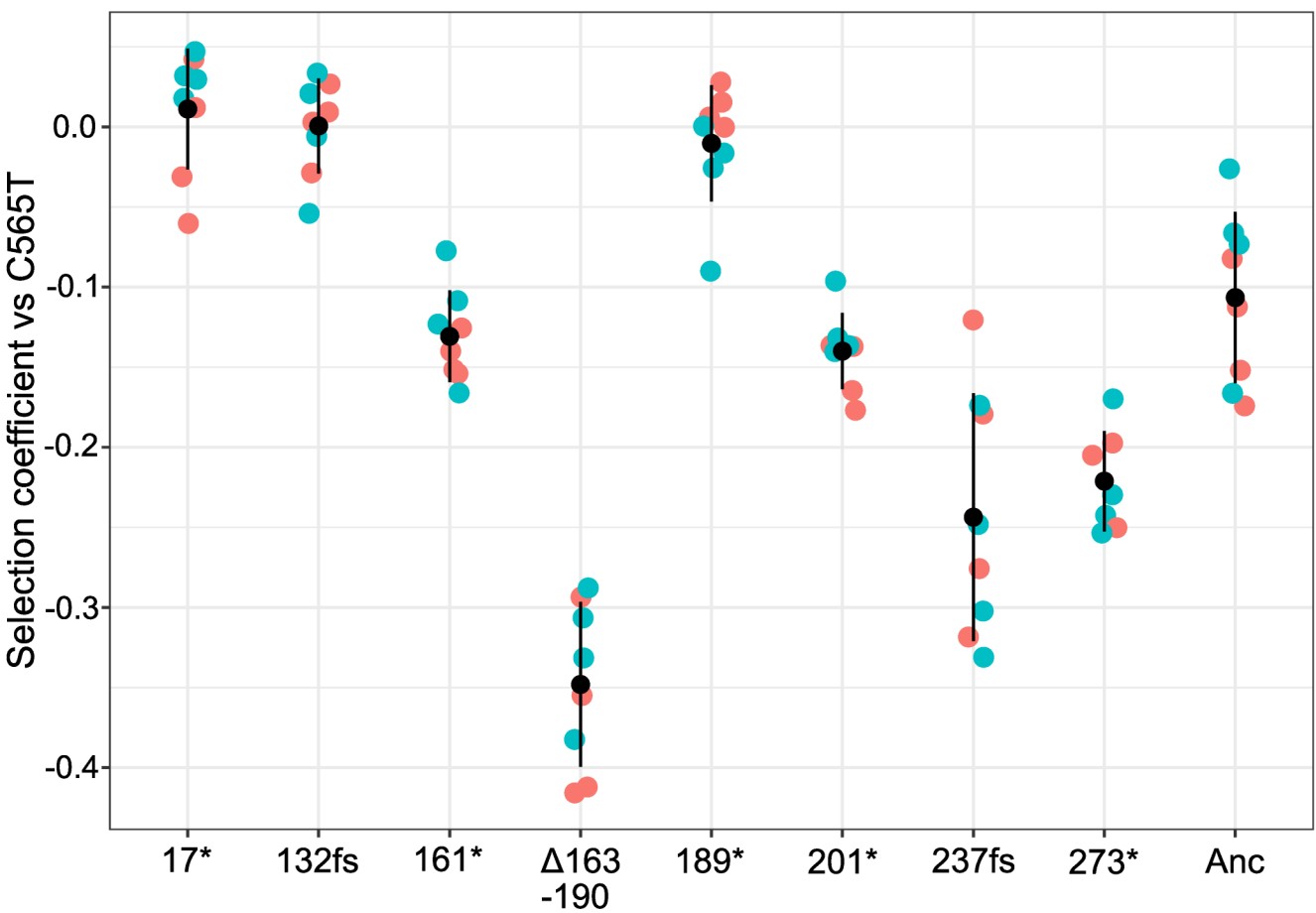

**Fig 2. Fitness of CC mutants containing mutations in *nlpD* relative to the C565T mutant.** Each *nlpD* mutation (labelled here with amino acid alteration) was reconstructed in a fluorescently labelled SBW25 Δ*wss* background and was competed against the C565T mutant (with initial ratio 1:1). A genotype with ancestral *nlpD* (Anc) was also included, showing a fitness benefit of the C565T mutant compared to the ancestor. Reciprocal pairwise competitions were used, with the indicated mutation in genotypes expressing either GFP (green) or mScarlet (red) fluorescence. Each competition involved 8 replicates, black dots represent the mean values and error bars are one standard deviation from the mean.

useful: any potential alteration of transcription by the C565T mutation may allow development of a mutation detection system and measurement of mutation rates.

Levels of transcript downstream from *rpoSp* were measured by RT-qPCR, using transcript directly upstream of *rpoSp* as a reference (Fig 3A). These measurements were performed using stationary phase cultures of WT *nlpD*, the C565T mutation and other *nlpD* loss-of-function mutations were reconstructed in SBW25 Δ*wss*. These additional mutations were included to assess whether the CC phenotype might result in alteration of transcription from *rpoSp*. Shaken cultures were used as opposed to static cultures to homogenously induce stationary phase. These assays revealed a ~7-fold increase in transcript produced across *nlpD* (Fig 3B), indicating the presence and expression of *rpoSp* in stationary phase cultures. Remarkably, while three mutations distal to *nlpD* had similar levels of transcript as in ancestral SBW25 Δ*wss*, the C565T mutation significantly increased the relative level of transcription across *rpoSp* by a further ~6.6-fold ((Fig 3B); p < 0.0001; one-way ANOVA, F4,15 = 109.5, p < 0.0001, pairwise difference assessed with Tukey HSD). This further transcriptional activation to *rpoS* may be responsible for the fitness advantage of the C565T mutant over at least 5 other loss-of-function mutants competed in static microcosms (Fig 2).

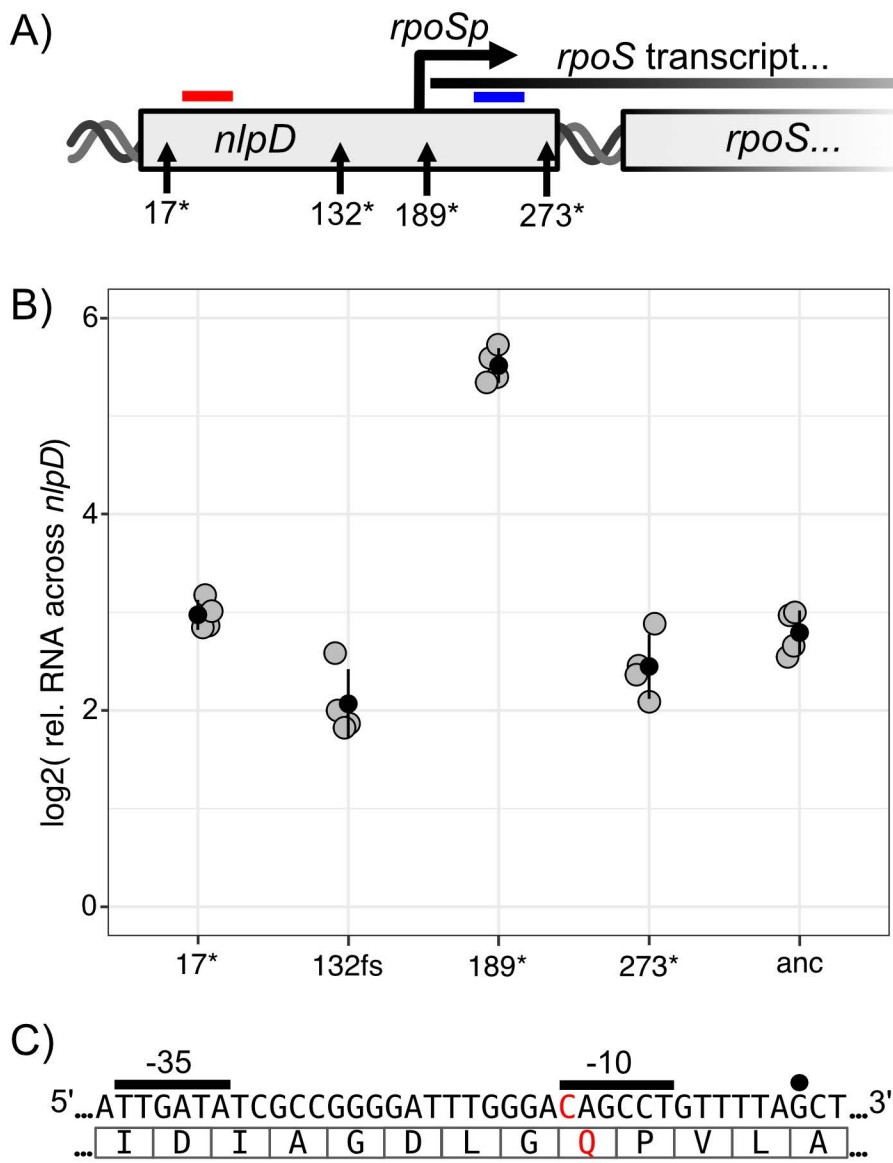

**Fig 3. The Q189 mutation elevates transcription from *rpoSp* compared to other *nlpD* mutants or the ancestral sequence.** A) The location of the four mutants, the regions with quantified transcript, and the predicted region of *rpoSp*. Transcript levels were measured downstream (blue bar) relative to upstream (red bar) of *rpoSp*. Transcription across *nlpD* was measured for mutants with loss-of-function mutations at codon 17 (TGA), 132 (a frameshift), 189 (TAG) or 273 (TAA), as well as the ancestral (anc) SBW25 Δ*wss* B) Measures of transcript across *nlpD* for the five *nlpD* genotypes. Strains were grown to stationary phase, mRNA was extracted, reverse transcribed and qPCR performed on regions up- and down stream of *rpoSp*. The plot depicts the relative amount of transcript downstream of *rpoSp* relative to upstream. The Q189* mutation causes a large (~42-fold) increase in transcript from *rpoSp*. Black dots represent the mean values of four biological replicates (gray dots) and error bars one standard deviation from the mean. C) The local sequence of the C565T mutation and predicted sigma factor binding regions. The nucleotide position of the C565T mutation is presented in red (with codon 189 translated into a glutamine as represented by the underlying red 'Q'). The mutation is located -12 bp from the modal position of the TSS for *rpoS* indicated by the overlying black point (also see S1 Fig and S2 Table). Overlying black lines mark presumed binding sites for the sigma factor RpoD, due to similarity to validated RpoD binding sites in E. coli [25].

Increased transcription across *nlpD* following the C565T mutation suggested that the mutation is proximal to *rpoSp*. We mapped the transcription start site (TSS) of *rpoSp* to confirm the approximate position of *rpoSp* in SBW25. Using 5'RACE, the beginning of the transcript from *rpoSp* (in both SBW25 and SBW25 Δ*wss*) was mapped to approximately 12 bp downstream of the C565T mutation (Fig 3C and S1 Fig and S2 Table). Transcription of *rpoSp* was previously mapped within one base-pair in *P. aeruginosa* [22]. Similar to *P. aeruginosa*, *rpoSp* has -10 region of high GC-content [22]. The C565T mutation results in a change in the -12 position from the TSS, resulting in a 'TAGCCT' -10 motif. This mutant motif matches the consensus -10 RpoD binding-site in *E. coli* for experimentally validated promoters ('TANNNT') [25], explaining the increased transcription caused by C565T. The spatial relationship of C565T to the TSS holds strong similarity to a previously described study [9], that identified high rates of T to C point mutations 7 bp outside the transcription start site of the transcriptionally induced gene *thyP3*.

Importantly, in this study, RT-qPCR and promoter mapping provide partial explanation for the parallel evolution of C565T: the mutation could activate *rpoSp* and cause an adaptive benefit over some *nlpD* loss-of-function mutants in selective static microcosms. However, this still does not readily explain the near deterministic manner of the mutation compared to all other *nlpD* loss-of-function mutants (Fig 2). We hypothesised that a high rate of the C565T mutation might present this mutation to selection, resulting in near deterministic parallel evolution. The dramatic increase in transcription from *rpoSp* allowed construction of a simple mutational reporter, such that the mutation rate of the C565T mutation could be measured without the confounding influence of changes to NlpD.

## The C565T mutation rate is orders of magnitude higher than the average C to T mutation rate

We designed and constructed a mutational reporter that would allow measurement of the frequency of C565T mutants in growing populations of SBW25. The construct consisted of a 401 bp portion of *nlpD* (with C565T central to this construct) fused to a kanamycin resistance gene. Upon acquisition of a C565T mutation, the transcription from *rpoSp* is expected to increase, causing cells to become resistant to kanamycin, thus allowing identification of rare C565T mutations by colony growth on agar plates supplemented with kanamycin. This '*rpoSp-kan*' selectable reporter was integrated into the genome of SBW25 at a site distal to *nlpD* thus allowing identification of C565T mutations without interfering with *nlpD* function.

This reporter was assessed for use in the context of fluctuations assays, which allows estimates of the C565T mutation rate. The fluctuation assay models that we used assume neutral expansion of the mutant clones in growing populations [26]. We performed fitness assays and measured a near-neutral effect of these C565T mutations (in the context of the *rpoSp-kan* construct) in growing cultures (S2 Fig). Fluctuation assay models also assume an equal likelihood of mutation per cell across growth phases (i.e., mutation rate is not "induced" at a simultaneous time point). Such an induction of mutation rate could conceivably result from increased *rpoSp* activity in stationary phase, and *rpoSp* was shown to be induced in fluctuation assay conditions using RT-qPCR (S3 Fig). Despite this induction, the occurrence of the C565T in the reporter occurred at similar magnitudes before and after cells entered stationary phase (S4 Fig) – though some significant reductions in mutational frequency were observed at some time points. These measures suggest fluctuation assays from early stationary phase samples (which we would use below) do not cause significant over, or under-estimation, of mutation rate.

Fluctuation assays were performed [27] on cultures of SBW25 harbouring the '*rpoSp-kan*' reporter. Twelve independent cultures were grown from a small inoculum (~1000 CFU mL$^{-1}$) in order to prevent inclusion of mutant strains in the initial inoculum. Assays were performed

in shaken media for 22 h of growth (until densities which induced *rpoSp* in preliminary experiments S3 Fig). Samples of the cultures were then plated on agar plates supplemented with high concentrations of kanamycin (400 mg L⁻¹) to select for C565T mutation in the reporter. To prevent mutants deriving from selective amplification of the construct [28], only colonies of sufficiently large size equivalent to plated controls with the C565T mutation were marked (see methods) and a subset of eight colonies were Sanger sequencing from each culture. The majority of sequenced colonies contained a C565T mutation in the reporter (60 of 96 – see S3 Table and raw data [29]), some colonies (6 of 96) had a A564G mutation, and the remaining colonies did not feature mutation in the reporter. This fraction of identified mutants was used to calculate the frequency of mutants in each culture of the fluctuation assay (S3 Table).

Under these conditions, the mean frequency of the C565T mutant in *rpoSp-kan* is one in $\sim 2 \times 10^5$ cells in the background of SBW25 (Fig 4A (left)). The mutation rate of the C565T mutation was $4.2 \times 10^{-7}$ per base-pair per replication (95% CI = $4.5 - 3.9 \times 10^{-7}$), as derived by a MSS Maximum Likelihood Method. In comparison, a mutation accumulation study estimated the average rate of C to T mutations in SBW25 at $7.4 \times 10^{-11}$ per base-pair per cell division [4]. This comparison is made less precise by the methodological differences and sources of statistical uncertainty between the assays. For instance, mutation accumulation assays rely on small numbers of mutations which may be subject to selective forces [30], and fluctuation assays are sensitive to violations of the assumptions of the Lea-Coulson model [26]. With these distinctions in mind, our measures of the C565T mutation rate broadly indicate a $10^3$ to $10^4$ order of magnitude difference between rates of C565T and the genomic average of C to T mutations. Interestingly, these fluctuation assays revealed a mutational hotspot slightly broader than base-pair 565 of *nlpD*. Sanger sequencing of mutant colonies identified 6 of 96

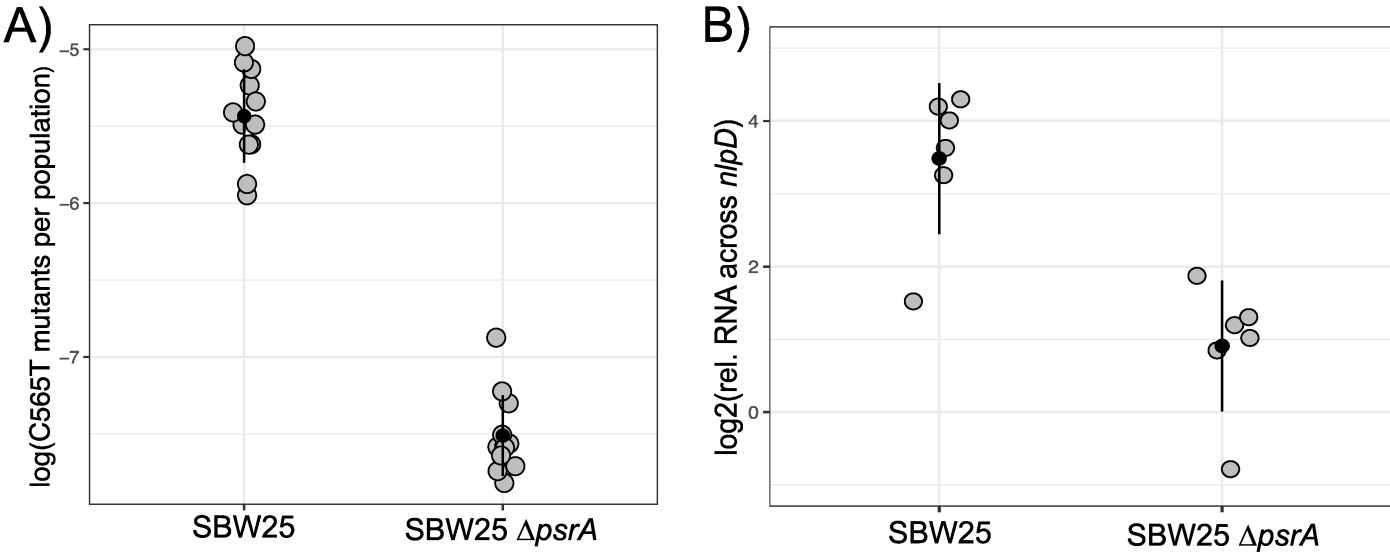

**Fig 4. The frequency of the C565T mutation is associated with induction of *rpoSp*.** A) The frequency of C565T mutants identified using a *rpoSp-kan* reporter in populations of SBW25 and SBW25 Δ*psrA*. SBW25 and SBW25 Δ*psrA* (both with the *rpoSp-kan* reporter were grown from a small inoculum (~1000 CFU mL⁻¹) for 22 h and samples plated on agar with and without selective kanamycin to measure the frequency of the C565T mutant (in the *rpoSp-kan* reporter). Data points represent measures of the frequency of the C565T mutant in a single culture. C565T mutants in the *rpoSp-kan* culture were confirmed via Sanger sequencing. Black dots represent the mean values of 12 biological replicates (grey dots) and error bars one standard deviation from the mean. B) Deletion of *psrA* reduces transcription from *rpoSp*. SBW25 and SBW25 Δ*psrA* were grown as per fluctuation assays and RT-qPCR was performed using methods described in Fig 3. SBW25 Δ*psrA* had ~4-fold reduction in promoter activity from *rpoSp*. Black dots represent the mean values of six biological replicates (gray dots) and error bars one standard deviation from the mean.

sequenced kanamycin-resistant colonies (from 4 independent cultures) of SBW25 *rpoSp-kan* having a A564G mutation in the reporter construct. Presumably this mutation can also activate transcription of *rpoSp* and occurs in *nlpD* in populations of SBW25, but is not selected in static microcosms because it results in synonymous mutation. In contrast, the high rate of the C565T mutation – when *rpoSp* is nested within *nlpD* – results in loss-of-function of NlpD, causing a high frequency of CC mutants in populations of SBW25.

## The high mutation rate is conditional on transcription from the *rpoSp*

Fluctuation assays identified high rates of the C565T mutation associated with *rpoSp*. Unknown is the underlying mechanism, and specifically whether transcription from *rpoSp* is the cause of the elevated mutation rate at base pair 565. We had identified the likely presence of *rpoSp* medial to *nlpD* using qPCR, and confirmed an increase in transcription from this region in stationary phase cultures (Fig 3B), but lacking was understanding of the functional relationship between *rpoSp* activity and the production of mutants.

Following the observation that the C565T mutation affected the -10 region of *rpoSp* and resulted in higher transcription, the transcriptional effects of the Q189W (sequence TGG) mutation on transcription from *rpoSp* were measured by RT-qPCR. The Q189W mutation was associated with a reduced mutational parallelism (Fig 1B). No measurable induction of expression from *rpoSp* was detected for cultures of SBW25 Δ*wss nlpD* Q189W (S5 Fig). This Q189W mutation also further removed one of the consensus nucleotides from the predicted -10 RpoD binding region [25]. These results provide further evidence that the -10 region of *rpoSp* is crucial for determining transcription from *rpoSp*, and supports the hypothesis that transcription from *rpoSp* causes the high C565T mutation rate.

A second region of *rpoSp* important for determining levels of transcription, and possibly rates of the C565T mutation, is the PsrA binding region. In *Pseudomonas*, transcription of *rpoS* is positively induced by the transcriptional regulator PsrA, which binds upstream of *rpoSp* and within *nlpD* [21] and likely initiates transcription following post-translational activation of PsrA [31]. The role of PsrA as an inducer of transcription from the SBW25 *rpoS* promoter was confirmed by quantitative RT-qPCR. Transcript from *rpoSp* was measured for cultures of SBW25 and SBW25 Δ*psrA* (Fig 4B). The absence of PsrA reduced transcription from *rpoSp* by ~6-fold, demonstrating that the level of transcription from *rpoSp* is conditional on PsrA.

Given the existence of *rpoSp* within *nlpD*, positive activation of *rpoSp* by PsrA, the position of base pair 565, and the general association between mutational parallelism and transcription from *rpoSp*, we reasoned that transcriptional activity at the *rpoS* promoter is necessary for hyper-mutation at base pair 565. This hypothesis draws on the finding of Sankar and colleagues, which showed that induction of transcription can elevate rates of a specific T to C mutation -7 bp to the TSS of *thyP3* [9]. To this end, we constructed a mutational reporter to assess the effect of PsrA on the rate of the C565T mutation by introduction of '*rpoSp-kan*' into the SBW25 Δ*psrA* background. As before, the neutral effect of these mutants were assessed for appropriate use in fluctuation assays [26]. The C565T mutations that arose in the SBW25 Δ*psrA* mutational reporter did not cause a fitness advantage in growing cultures (S2 Fig). We then performed fluctuation assays identical to those with SBW25. The frequency of C565T mutants in culture was two orders of magnitude lower when transcription from *rpoSp* was reduced by deletion of PsrA (Fig 4A) right**).** The mutation rate of the C565T mutation in the background of SBW25 Δ*psrA* was $7.2 \times 10^{-9}$ per base-pair per replication (95% CI = $9.2 - 5.4 \times 10^{-9}$), as derived by a MSS Maximum Likelihood Method. This represents a ~60-fold decrease compared to the rate of mutation in SBW25; the ~5000-fold elevated rate of the C565T

mutation is reduced to ~80-fold. Together, these measures demonstrate the high C565T mutation rate is conditional on *rpoSp* and the induction of transcription by PsrA.

## Discussion

The work described here shows the open reading frame of *nlpD* features a nucleotide at position 565 (codon 189) that mutates from cytosine to thymine at a rate orders-of-magnitude above the average per base-pair mutation rate for *P. fluorescens* SBW25 [4]. This high mutation rate was initially suspected following observations of near-deterministic levels of parallel evolution caused by the C565T mutation. Analysis of the fitness effects of a range of *nlpD* mutants showed that elevated fitness benefits due to the C565T mutation are insufficient to explain its frequent occurrence. Genetic analysis showed that C565T occurs within the promoter sequence of the stationary phase sigma factor RpoS. The C565T mutation has two phenotypic effects: it causes cell chaining while concomitantly enhancing transcription from *rpoSp*. This elevation in transcription allowed construction of a mutational reporter permitting measures of the C565T mutation rate. Finally, we show that this extreme mutation rate is dependent on induction of transcription by the positive inducer PsrA.

A rare feature of this study is the ability to determine a specific rate of mutation which fuels parallel evolution. Instances of parallel evolution caused by elevated mutation rates are both predicted by theory [15] and are supported by numerous experimental studies (see references within [5]). However, rarely are hotspots associated with a selectable phenotypic change that allows determination of specific substitutions rates [9,32–34]. In the case of C565T, an estimate of the mutation rate was made possible following the fortuitous identification that the mutation increased transcription from *rpoSp*. This observation allowed construction of the mutational reporter, which specifically reports the occurrence of the C565T mutation in *rpoSp* outside of the context of *nlpD*. The specificity of the reporter prevents assessment of mutation rate across *nlpD* (or the local region) and the contribution of broader mutagenetic processes such as transcription-associated mutagenesis remains unknown [35]. The necessary disentangling of *rpoSp* from *nlpD* further provides uncertainty in the true mutation rate of the C565T mutation in the context of *nlpD*, leaving indirect the association between the measured mutation rate and the parallelism of the C565T mutation. Regardless of these limitations, the reporter construct demonstrates that the naturally occurring promoter in *rpoSp* features a mutational hotspot that is dependent on the presence of PsrA. We are not aware of other reports of a mutational hotspot occurring within a wild-type promoter.

The single study that bears greatest similarity to the results reported here is that by Sankar et al [9]. The authors, using the model species *Bacillus subtilis*, reported thymine to cytosine mutations occurring at a rate of ~$2.7 \times 10^{-8}$ per cell per generation, specifically -7 to the transcription start site within the IPTG-inducible promoter of *thyP3*, resulting in promoter malfunction. In comparison, we identified cytosine to thymine mutations at a rate of ~$4.2 \times 10^{-7}$ per cell per replication, -12 to the transcription start site in a positively induced promoter, resulting in promoter gain-of-function (and concomitant NlpD loss-of-function). These similarities in position and magnitude suggest a common mutational process. Sankar and colleagues proposed solvents capable of causing deamination, which would cause nucleotide lesions on the template strand in the transcriptional bubble. Another possibility is that sigma-factor binding and holoenzyme assembly act as a barrier to mutational surveillance such as mismatch repair or the repair of nucleotide lesions. In the case of the C565T mutational hotspot, mutagenesis is influenced by PsrA, which is required for the high C565T mutation rate. PsrA is predicted to bind to the PsrA binding site approximately (50 to 29 bp upstream of the C565T mutation), as informed by DNAse footprint assays in *P. putida* which share an

almost identical binding site sequence [21]. PsrA binding at *rpoSp* provides an interaction not present in the experimental promoter system used by Sankar and colleagues. This positive inducer may exacerbate the hotspot by preventing activity of mutational repair proteins [36]. Alternatively, PsrA may 'hold' the *rpoSp* transcriptional machinery in place until PsrA is post-translationally activated, resulting in solvent attack [23,31]. Disentangling the role of these two processes may prove complicated from a genetics approach, but developing strategies that position nucleotides within the holoenzyme [37] may provide clues as to why C565T is particularly prone to mutation. Whatever the exact cause, we predict that similar phenomena may occur elsewhere, and may be detected by direct assessment of mutation rates in promoters with positive transcriptional regulation – possibly by taking advantage of techniques such as maximum-depth sequencing [38].

Positive regulation of promoters may elevate local mutation rates, potentially accelerating adaptive evolution. If mutational hotspots can be identified at other inducible promoters, they might uniquely contribute to the supply of genetic variation required for adaptation. The spatial specificity of these hotspots is likely to restrict mutations to transcriptionally critical regions, suggesting a mechanism that biases mutations toward functional outcomes. Notably, these hotspots may be particularly active when promoters are induced, implying their activity is heightened in promoters driving environmentally responsive genes. This stands to establish a scenario where bacteria, in response to environmental cues, experience elevated mutation rates in promoters, generating variability in niche-relevant transcript levels. Such promoter mutations would likely arise and be exposed to selection at significantly higher rates compared to functional changes in coding regions. Moreover, these mutations might even revert rapidly when high transcript levels are no longer advantageous. This dynamic could explain experimental findings in which a high proportion of beneficial mutations in bacterial populations are promoter mutations [39,40].

The C565T mutational hotspot is striking because of its location within the highly conserved *nlpD-rpoS* pseudo-operon, a genetic arrangement shared across Gammaproteobacteria [41]. While our focus has been on the elevated mutation rate at *rpoSp*, the broader ecological significance of this hotspot remains unexplored, both in *P. fluorescens* and other Gammaproteobacteria. The mutation has two primary effects: truncation of NlpD, resulting in cell chaining, and increased transcription of *rpoS*. Although translational and post-translational controls may moderate the effects of higher *rpoS* transcript levels [41], the phenotypic impact of NlpD truncation is clear. In laboratory microcosms, cell chaining enhances adaptation to static, nutrient-rich environments [17]. However, in natural ecosystems, this phenotype might carry trade-offs: while chains may facilitate biofilm formation or nutrient capture, shared periplasmic spaces could amplify vulnerability to phage infection or lysis, jeopardizing entire chains. The presence of a mutational hotspot in *rpoSp*, despite its potential morphological costs, raises questions about its evolutionary persistence in a conserved genetic region. If this hotspot is widespread in Gammaproteobacteria, it suggests a yet-unknown selective advantage linked to the nesting of *rpoSp* within *nlpD*. Uncovering the mechanisms driving this arrangement promises to deepen our understanding of how complex genetic architectures are maintained over vast evolutionary timescales.

## Methods

### Media and strains

Derivatives of *Pseudomonas fluorescens* SBW25 were used in all experiments [42]. The SBW25 Δ*wss* strain is identical to that previously described [17]. *Escherichia coli* strains were used for the alteration of the SBW25 genome - *E. coli* Top10, *E. coli* DH5-α λ*pir,* *E. coli* pRK2013, *E. coli*

S17-1 $\lambda_{pir}$. All strains were stored in ~25% glycerol saline solution at -80 °C. Unless otherwise stated, *P. fluorescens* was cultured using King's B (KB) medium [43] and *E. coli* using lysogeny broth (LB) medium [44]. Media was solidified using 1.5% agar. Unless otherwise stated, liquid cultures were incubated with orbital shaking (220 rpm) at either 28 °C for ~48 h (for *P. fluorescens*) or 37 °C for 24 h (for *E. coli*). Co-cultures of *E. coli* and *P. fluorescens* were incubated at 28 °C. Media was supplemented with the following selection agents for the construction of strains: kanamycin (50 mg L⁻¹), nitrofurantoin (1 g L⁻¹), tetracycline (15 mg L⁻¹). Kanamycin was added to LB agar at 400 mg L⁻¹ for selection of C565T mutants using the 'rpoSp-kan' cassette (see below). Double recombinants from two-step allelic exchange using our in-house pUI*sacB* plasmid were selected in TSY10 media [45].

## Reconstruction of mutations and insertion of genes

The introduction of plasmids into *nlpD* loss-of-function strains failed after multiple attempts – it is important to note that complementation or reconstruction of *nlpD* mutations requires transfer of those *nlpD* mutations into other genotypic backgrounds to make desired constructs.

To enable fitness assays, derivatives of SBW25 were marked with *sgfp2* and *mScarlet-I* at the neutral *attTn7* locus [46] using plasmids described in detail elsewhere [47]. *E. coli* S17-1 $\lambda_{pir}$ hosting plasmids pMRE-Tn*7*-152 or pMRE-Tn*7*-155 were cultured overnight with selective antibiotics along with target SBW25 genotypes. Bi-parental mating was then performed on the resulting cultures (see below). Transconjugants of *P. fluorescens* were selected on LB agar containing 0.1% W/V arabinose, nitrofurantoin and kanamycin to select for transposition of the mobile section of pMRE-Tn*7*-152 or pMRE-Tn*7*-155 into the *attTn7* site. Resulting fluorescent colonies were then streaked on LB agar containing kanamycin, and resulting colonies were assessed by PCR for integration at the *attTn7* site. Overnights from these colonies were made with selective kanamycin and stored at -80 °C.

Reconstruction of mutations, or the deletion of genes, was achieved using two-step allelic exchange using sucrose counter-selection [48]. To reconstruct existing mutations in other genotypes, template containing the mutation of interest was amplified with primers approximately 500 - 800 up- and downstream of the mutation targeted for reconstruction (i.e., approximately 1400 bp in total) using 'Q5 High-Fidelity DNA Polymerase' (NEB). In the case of reconstruction of *nlpD* mutations evolved from Q189W, the Q189W mutation was reverted to the WT sequence by strand overlap extension PCR [49]. To delete genes, similar sized regions either size of the gene of interest were amplified, using primers with complementary tails at one end of each fragment. These fragments were cloned using an inhouse plasmid 'pUI*sacB*' (European Nucleotide Archive (ENA) accession number OZ219427), which features genes for tetracycline resistance (*tetA*), fluorescence (*msfgfp*) and sucrose counter selection (*sacB*). pUI*sacB* was linearized using restriction enzymes and then amplified using 'Q5 High-Fidelity DNA Polymerase'. Amplified pUI*sacB* was digested with DpnI. The *nlpD* fragments were amplified with primers with a 5' tail complementary to 20 nucleotides to the ends of the pUI*sacB* amplicon. Both amplicons were purified using a 'Qiaquick PCR purification kit' (Qiagen), product concentrations were quantified, and then joined into a circularized plasmid by 'NEBuilder HiFi DNA assembly' (NEB) following manufacturers protocols. Assembled products were transformed into chemically competent *E. coli* Top10 cells, with transformed cells plated on LB agar containing tetracycline. Several colonies from each transformation were subject to colony PCR to confirm the correct sized insert and several colonies were used to inoculate overnight cultures. Overnight cultures were also started of *E. coli* pRK2013 (using LB supplemented with kanamycin), and the target host strain of the

reconstruction (SBW25 Δ*wss*). Resulting *E. coli* pUI*sacB-nlpD* cultures were then frozen, and the pUI*sacB-nlpD* was transferred to the target host by tri-parental mating (see below), with transconjugants selected with LB agar plates supplemented with tetracycline and nitrofurantoin. Several resulting colonies were streaked on TSY10 plates to counter-select for pUI*sacB* recombinants. Resulting colonies were scanned for the phenotypic effects of the reconstructed mutation – in this case cell chaining, by visual identification using microscopy – and colonies were streaked on KB agar plates, along with LB-tet plates to confirm loss of pUI*sacB*. Double recombinant mutant colonies were then cultured in shaking KB microcosms prior to storage. PCR and Sanger sequencing was performed on all reconstructed mutants to ensure the correct mutation (and the absence of additional mutations) at the intended locus.

Sucrose counter selection was not used for construction of SBW25 Δ*wss nlpD* C565T A566G, or for reconstruction of *nlpD* mutations into SBW25 Δ*wss* (used as template for subsequent reconstruction using pUI*sacB*), rather, we used a previously described two-step allelic exchange method [50,51]. Briefly, forward and reverse primers with binding sites ~800 bp from the C565T A566G mutation were used to amplify the template DNA. This fragment was ligated into the tetracycline-resistant pUIC3 plasmid [51] and cloned in *E. coli* DH5-α λ$_{pir}$. Conjugation with *E. coli* pRK2013 was used to transfer the pUIC3 construct into SBW25 Δ*wss*, and single recombinants were selected for with tetracycline and nitrofurantoin. Recombinant clones were used to initiate cultures which were treated with cycloserine with counter-selected single recombinants. The culture was plated and single colonies were stored which were cured of the pUIC3 plasmid, and the absence of mutational scars and the presence of the desired mutation in *nlpD* was confirmed by Sanger sequencing.

Construction of the '*rpoSp-kan* reporter' and fluorescent strains with this reporter were also constructed using 'NEBuilder HiFi DNA assembly' (NEB). To construct the '*rpoSp-kan* reporter', 401 bp of *nlpD* (with the C565T mutation situated centrally) was amplified along with a kanamycin resistance gene (referred to here with standard name '*kanR*', specifically encoding Aminoglycoside phosphotransferase type III) and a backbone plasmid (extracted from MPB15151, see [52]) encoding Tn*7* elements, an origin of replication and a tetracycline resistance enzyme. These three fragments were assembled using 'NEBuilder HiFi DNA assembly Master Mix' (NEB) (see above), and transformed into top10 chemically competent cells. The resulting plasmid (pTn*7-rpoSp-kan*, ENA accession number OZ219428) was extracted from liquid culture via miniprep using a 'QIAprep Spin Miniprep Kit' (Qiagen) transformed along with the pUX-BF13 (to express transposition machinery, [53] into derivatives of SBW25 using electroporation (protocol previously described [46]). Resulting transformants were selected using LB agar supplemented with tetracycline and the insertion of the construct at the *attTn7* was confirmed by PCR and electrophoresis. The integrated *rpoSp-kan* genetic construct validated with Sanger sequencing. The Tn*7-rpoSp-kan* reporter plasmid was similarly modified to express either sGFP2 or mScarlet-I (resulting in plasmids pTn*7-gfp-rpoSp-kan*, or pTn*7-scarlet-rpoSp-kan*, ENA accession numbers OZ219426 and OZ219429 respectively), and this construct was integrated into the *attTn7* region of SBW25 to enable fitness assays. Attempts to introduce a C565T mutation into pTn*7-rpoSp-kan* failed over multiple attempts, and spontaneous mutants were used to derive SBW25 *attTn7*::*rpoSp*C565T-*kan*, SBW25 *attTn7*::gfp-*rpoSp*C565T-*kan* and SBW25 *attTn7*::scarlet-*rpoSp*C565T-*kan* (also in backgrounds without *psrA*).

Transfer of plasmids from *E. coli* cloning strains to derivatives of SBW25 was performed with tri-parental or bi-parental mating or by electroporation where specified. Triparental mating required overnight cultures of the donor strain with the intended plasmid, a helper strain (*E. coli* pRK2013) and the target *Pseudomonas* host strain. Where the donor was *E. coli* S17-1 λ$_{pir}$, the helper strain was omitted. When the cultures were grown to saturation, 1 mL samples

of SBW25 culture were heat shocked at 45 °C for 20 mins. Meanwhile, 500 μL of *E. coli* culture was washed of remaining selective antibiotics by centrifugation, and resuspension of the cells in 500 μL of LB. The heat-shocked SBW25 culture was then added to the *E. coli* culture, and the mixed culture was centrifuged, supernatant was removed and the cells resuspended in 100 μL of LB. The resuspension of cells was then spread over a ~4 cm diameter region of a plate of LB agar and incubated for ~24 h. Transconjugants were then resuspended in 1 mL of LB and dilutions were plated on selective LB agar plates as indicated below to select for transconjugants.

## Experimental evolution of the C565T mutants

The suite of *nlpD* mutants evolved from SBW25 Δ*wss* and SBW25 Δ*wss nlpD* C565T A566G were evolved using previously described methods (see [17]). Extra care was taken to limit contamination with a potential C565T mutation in the SBW25 Δ*wss* inoculum. Accordingly, individual colonies of SBW25 Δ*wss* were used to inoculate individual KB microcosms, which were then incubated statically for five days. The genotype SBW25 Δ*wss nlpD* C565T A566G was inoculated direct from an isogenic frozen stock into replicate microcosms of KB media. These microcosms were incubated for six days and a sample of the microcosm was stored at -80 °C, and dilutions of the microcosms were plated over several KB plates. Resulting colonies were assessed by eye for ancestral appearing morphologies with subtle alteration in opaqueness, and different morphotypes were assessed by microscopy for a chaining cellular phenotype. Individual mutant colonies were then streaked on KB agar plates to produce single colonies which were used to produce cultures from which *nlpD* was Sanger sequenced.

## Fitness assay of *nlpD* mutants

Fitness assays of *nlpD* mutants – or *rpoSp-kan* constructs – were performed as previously described [39], with the following exceptions. For fitness assays of the '*rpoSp-kan*' mutants, cultures were prepared to replicate fluctuation assays, with 24 h incubated shaking cultures of each competitor mixed (50:50), sampled for flow cytometry and plated on KB agar, and the competitors diluted $10^{-7}$ in new KB microcosms to found the test microcosms. These test microcosms were then sampled again (with flow cytometry and plating on KB agar) after 22 h of shaking at 220 rpm. Fitness assays compared strains marked with sGFP2 or M-Scarlet-I [47]), with fluorescent proteins swapped for half the replicates to correct for fitness cost of the fluorophores. Fitness assays of *nlpD* mutants competed in 24 h statically incubated microcosms also included an equal ratio of non-marked ancestral to help recreate the conditions in which *nlpD* mutants evolved. The relative frequency of either fluorescent marked mutant was measured using a 'MACSQuant VYB' flow cytometer (Miltenyi). Samples were run with a 'low' flow rate, 'moderate' mixing and 'standard' cleaning between samples. Samples were prepared for flow cytometry diluted with filtered PBS buffer in 96-well plate wells in order to achieve ~500-800 recorded events per second. The cytometer had a trigger set at value 1.4 of the SSC channel, and events were gated to not include events from the filtered PBS buffer used to dilute samples, and at least 20,000 events were sampled. Events were plotted of events from 525/50 nm (sGFP2) and 615/20 nm (M-Scarlet-I) detectors. We calculated selection coefficients per generation with the regression model $s = [\ln(R(t)/R(0))]/[t]$, in which R is the ratio of each competitor at tested time points, and t is the number of generations [54]. Fitness assays were performed with 8 replicates performed over 2 separate occasions.

## Rapid amplification of cDNA ends

Triplicate cultures of SBW25 and SBW25 Δ*wss* were each inoculated with a single colony and incubated statically for 24 h. Samples of SBW25 and SBW25 Δ*wss* (200 μL and 400 μL

respectively) were taken, and total RNA was extracted following manufacturer's instructions using an 'RNeasy Mini Kit' (Qiagen) in conjunction with 'RNAprotect tissue reagent' (Qiagen). The 5' end of *rpoS* mRNA was then mapped using a '5' RACE System for Rapid Amplification of cDNA, version 2.0' (Invitrogen). A sample of 1 μg of total RNA as the sample input, with the first extension primer directly outside of the *nlpD* open reading frame. PCR products were amplified over two rounds, and resulting cDNA was amplified with '*Taq DNA Polymerase*' (NEB) to allow cloning of the resulting products into pCR2.1 using a 'TOPO TA Cloning Kit' (Invitrogen), with plasmids transformed into top10 chemically competent cells. At least 10 resulting colonies were stored and colony PCR was used to amplify the region of insert and an insert-annealing primer used for a Sanger sequencing reaction to identify the 5' start of the mRNA transcript.

## Quantitative RT-PCR

Quantification of transcript levels across *nlpD* was performed using qPCR using previously described methods [39]. Briefly, total RNA was isolated from shaken cultures grown either to specified time points or to stationary phase ($OD_{600}$ = 2.5 to 3.0). RNA was isolated using the 'SV Total RNA isolation system' (Promega). Cultures were centrifuged, and pellets were resuspended in TE buffer (10 mM Tris, 1 mM EDTA pH 8.0) with 0.4 mg mL$^{-1}$ lysozyme. The recommended protocol was then performed for extracting RNA from gram negative bacteria. Total RNA was then reverse transcribed using to DNA using a 'High capacity cDNA reverse transcription kit' (Applied biosystems). Resulting cDNA was then diluted 1:40 before use as template in the qPCR reaction (8 μL of diluted template per 20 μL reaction) using 'PowerUp SYBR Green Master Mix' (Applied biosystems). Reactions were performed in 'MicroAmp Optical 96-Well Reaction Plates' (Applied biosystems) and reactions were measured in a 'Quantstudio 1 Real-Time PCR System' (Applied biosystems). The change in transcript downstream of *rpoSp* was made relative to transcript upstream of *rpoSp*, but within *nlpD*, using the ΔΔCq method [55]. Two replicate qPCR measures were made of each biological replicate, with each run performed on a separate occasion.

## Fluctuation assays

In order to measure the rate of mutation of the C565T mutation, a construct (*rpoSp-kan*, see methods above) was employed which would allow selection of the mutation in *rpoSp* while removing *rpoSp* from the fitness-altering context of *nlpD*. Six individual transformants of the *rpoSp-kan* construct in both SBW25 and SBW25 Δ*psrA* were used for fluctuation assays, each used on two separate occasions. For each replicate measure, individual colonies of the reporter strain were grown (without antibiotic selection) and then inoculated into shaking KB cultures for 24 h. Cultures were diluted $1 \times 10^{-7}$ into 6 mL microcosms of KB media, and this culture was immediately plated to derive estimates of the inoculum (typically ~1000 CFU mL$^{-1}$). Cultures were then shaken for 22 h, and then diluted $1 \times 10^{-6}$ and plated on KB plates to get measures of CFU. Cultures of SBW25 were diluted 100 and 1000-fold for selective plating on multiple LB plates with kanamycin (400 mg L$^{-1}$), while SBW25 Δ*psrA* mutants were plated on selective plates without dilution. Duplicate control genotypes (cultured the same as the above) which already feature the C565T mutants were diluted $1 \times 10^{-6}$ or $1 \times 10^{-7}$ and plated on selective media. All plates were incubated for ~72 h at 28 °C. The total number of cells per plated culture were estimated from CFUs on KB media, while selective plates were visually scanned for colonies with a similar size to the control plates. To minimise the possibility of spontaneous mutants arising during slow growth on the selective plates, a minimum size was required for candidate colonies as compared with control colonies with C565T mutations in

the reporter (3 mm or 4 mm was the minimum diameter size on plates with respectively more or less than 100 visible colonies). Candidate colonies were the marked and counted, and a random selection of 8 such colonies were selected for Sanger sequencing. Upon identification of the C565T mutation in the construct, the total number of candidate colonies was corrected by the fraction of confirmed C565T mutants to produce an estimate of the true number of C565T mutants in the population. MSS maximum likelihood methods were used to analyse mutation rates using 'FALCOR' (https://lianglab.brocku.ca/FALCOR/; [56]).

Measures of the frequency over time (S4 Fig) of the C565T mutation in the *rpoSp-kan* reporter were acquired using similar methods as the above fluctuation assays. Exceptions were that the final fluctuation assay cultures were then grown for approximately 14.5 to 22 h with destructive sampling (each culture was not resampled) every ~90 minutes. Samples were diluted as appropriate to derive both total CFU concentration and for selective plating, which required multiple dilutions. Selective plating used eight LB agar plates supplemented with kanamycin (400 mg L⁻¹) for each dilution. Colonies were again incubated for ~72 h at 28 °C. Candidate mutants were selected due to a minimum colony diameter in reference to plates with control C565T mutants. Minimum size of colonies for selection was 3.5 or 4 mm, conditional on respectively more or less than 100 colonies being visibly present on the plates. Again, eight colonies were randomly selected per time point for Sanger sequencing to establish the true number for C565T mutants in the populations. If no mutants were selected (which happened on one occurrence), a further 24 colonies were randomly selected for sequencing. Experiments were performed over two separate occasions, with at least two independent replicate cultures sampled of each time point per occasion.

## Data visualisation

Data analysis was performed in 'Rstudio' [57] version 2023.9.1.494, utilising 'R' [58] version 3.1.2, and data visualisation was performed using packages 'ggplot2' [59]. Raw data and calculations performed to produce visualisation of data are available (see the Data Availability Statement).

## Supporting information

**S1 Fig. The position of the C565T mutation relative to the transcriptional start site (TSS).** The TSS of *rpoS* is approximately 12 bp downstream of the position of the C565T mutation (Red C). Circles represent the TSS from individually cloned reverse-transcription fragments from RNA expressed in stationary phase cultures of either SBW25 or SBW25 Δ*wss* (see S1 Table for all sequenced TSS positions). Cloned fragments were derived from one of three replicate cultures (indicated by either white, grey or black circles). The position of the mapped TSS of *rpoS* in *P. aeruginosa* is marked blue [22]. Triangles represent the consensus 'gearbox' promoter motif [60]. (PDF)

**S2 Fig. Relative fitness of SBW25 genotypes with C565T mutated *rpoSp-kan* reporters vs non-mutant reporter genotypes.** The *rpoSp-kan* reporter construct was made to express Green or mScarlet fluorescent proteins, and evolved in genotypes SBW25 or SBW25 Δ*psrA* to feature a C565T mutation in the *rpoSp-kan* reporter. Presented is the relative fitness of genotypes with C565T *rpoSp-kan* vs unmutated *rpoSp-kan*. Competitions were initiated with a 1:1 ratio and grown in a manner similar to fluctuation assays. Reciprocal pairwise competitions were used, with the C565T mutation in either a GFP (green) or mScarlet (red) background. A significant lower fitness was measured for the C565T mutant in *rpoSp-kan* in both backgrounds (Wilcoxon signed rank exact test of either SBW25 or SBW25 Δ*psrA*, V = 29, p-value

= 0.04431). Each competition involved 16 replicates, black dots represent the mean values and error bars one standard deviation from the mean.
(PDF)

**S3 Fig. Growth of SBW25 causes induction of transcription from *rpoSp*.** SBW25 was grown from a small inoculum (~1000 CFU mL$^{-1}$) for 14, 18 and 22 hrs and samples were used to measure CFU concentrations and processed (as per Fig 3) to measure induction of transcription from *rpoSp* relative to a region in *nlpD* upstream of *rpoSp*. Similar levels of activity from *rpoSp* were measured after 18 and 22 hrs of growth, and fluctuation assays used cultures grown for 22 hrs to ensure cells were in stationary phase.
(PDF)

**S4 Fig. The frequency of C565T mutant reporter strains as cultures approach stationary phase.** Fluctuation assays were performed as per Fig 3, except sampling occurred at earlier time points (four replicates per grouped time point). Samples at earlier time points ~15.8 hrs and ~17.3 hrs had significantly lower frequencies of the C565T mutation in the *rpoSp-kan* construct compared to the final time point of ~21.8 hrs (p = 0.04 and p = 0.03 respectively; one-way ANOVA, F(5,8.301) = 3.874, p = 0.04, pairwise difference assessed with Tukey HSD).
(PDF)

**S5 Fig. The Q189W mutation prevents transcription from *rpoSp* for cells in stationary phase.** SBW25 Δ*wss* (*nlpD* anc) and SBW25 Δ*wss nlpD*Q189W (which has a TGG sequence for *nlpD* codon 189 and a likely alteration of the *rpoSp* sequence) were grown to stationary phase, and mRNA was extracted, reverse transcribed and the regions up and down stream of *rpoSp* were used as templates for qPCR. SBW25 Δ*wss* has levels consistent with other measures of induced transcription from *rpoSp*, while the C565T A566G mutation to *rpoSp* removes any indication of transcription from *rpoSp* (i.e. there is no difference in the levels of transcript downstream compared to upstream of *rpoSp*). Black dots represent the mean values of five biological replicates (gray dots) and error bars one standard deviation from the mean.
(PDF)

**S1 Table. Mutational positions and residue changes of *nlpD* mutants arising from populations of SBW25 Δ*wss nlpD* (Q189W) grown in statically incubated microcosms.**
(DOCX)

**S2 Table. Positioning of the transcriptional start site (TSS) from total RNA extracts of SBW25 and SBW25 Δ*wss* using 5-prime RACE.**
(DOCX)

**S3 Table. Data used in calculating the C565T mutation rate using SBW25 *attTn7::rpoSp-kan* (genotype 'WT *r-k*' below) and SBW25 Δ*psrA attTn7::rpoSp-kan* (genotype 'Δ*psrA r-k*').**
(DOCX)

## Acknowledgments

Thanks to David Rogers and Ellen McConnell for advice during molecular cloning, Elisa Brambilla for laboratory management and the MPI for Evolutionary Biology sequencing team.

## Author contributions

**Conceptualization:** Andrew D. Farr, Peter A. Lind, Paul B. Rainey.

**Data curation:** Andrew D. Farr.

**Formal analysis:** Andrew D. Farr.

**Funding acquisition:** Paul B. Rainey.

**Investigation:** Andrew D. Farr, Christina Vasileiou.

**Supervision:** Peter A. Lind, Paul B. Rainey.

**Visualization:** Andrew D. Farr.

**Writing – original draft:** Andrew D. Farr.

**Writing – review & editing:** Andrew D. Farr, Peter A. Lind, Paul B. Rainey.

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
