## [Decision Letter · Decision Letter 0]

4 Oct 2024

Dear Dr Farr,

Thank you very much for submitting your Research Article entitled 'An extreme mutational hotspot in nlpD depends on transcriptional induction of rpoS' to PLOS Genetics.

The manuscript was fully evaluated at the editorial level and by independent peer reviewers. 

All reviewers acknowledged the manuscript’s rigor and its relevance to understanding mutational hotspots, but raised some substantial concerns about the current manuscript. They expressed concerns regarding overstated conclusions and methodological limitations, particularly in relation to the reported increase in mutation rates and the role of selection. They requested clarification, a more nuanced discussion, and additional experiments to enhance the mechanistic insights, along with minor textual adjustments.

**Reviewer #1** raised questions about the reported 5,700-fold increase in mutation rates. They suggested that there may be potential issues with the fluctuation assay and FALCOR analysis, particularly due to violations of the assumptions underlying the Lea-Coulson model. **Reviewer #2** expressed skepticism regarding the assertion that selection plays no role in genetic parallelism. They recommended including a discussion on the interplay between mutation and selection. Additionally, they highlighted limitations in the mutation rate assay and requested a clearer explanation of the mechanisms underlying parallelism, as well as addressing concerns about the variability in fitness data. **Reviewer #3** proposed investigating transcription-coupled repair pathways as an alternative explanation for the observed hotspots, and recommended comparing mutations in mismatch repair-deficient strains. They also recommended expanding the section on Kan selection results, providing more details on how mutation rates were derived.

Based on the reviews, we will not be able to accept this version of the manuscript, but we would be willing to review a much-revised version. We cannot, of course, promise publication at that time.

If you decide to revise the manuscript for further consideration at PLOS Genetics, please aim to resubmit within the next 60 days, unless it will take extra time to address the concerns of the reviewers, in which case we would appreciate an expected resubmission date by email to plosgenetics@plos.org.

If present, accompanying reviewer attachments are included with this email; please notify the journal office if any appear to be missing. They will also be available for download from the link below. You can use this link to log into the system when you are ready to submit a revised version, having first consulted our Submission Checklist .

PLOS has incorporated Similarity Check , powered by iThenticate, into its journal-wide submission system in order to screen submitted content for originality before publication. Each PLOS journal undertakes screening on a proportion of submitted articles. You will be contacted if needed following the screening process.

To resubmit, log into your Editorial Manager account and select the option 'Revise Submission' in the 'Submissions Needing Revision' folder.

We are sorry that we cannot be more positive about your manuscript at this stage. Please do not hesitate to contact us if you have any concerns or questions.

Yours sincerely,

Ivan Matic

Academic Editor

PLOS Genetics

Lotte Søgaard-Andersen

Section Editor

PLOS Genetics

Reviewer's Responses to Questions

**Comments to the Authors:**

Reviewer #1: I would firstly like to applaud the authors on the manuscript, which includes a series of rigorous experiments, is clearly written and presented, and presents interesting results that further our knowledge of mutational hotspots in bacteria. The primary result of this work is that a C->T mutational hotspot found at nucleotide position 565 in the gene nlpD is driven by transcription. The authors show that nlpD C565 is additionally situated within the promoter region of the neighbouring stress response gene rpoS. They report a positive correlation between induction of the rpoS promoter region and rates of C565T mutation. They then use these results to estimate an increase in mutation rate of ~5700-fold over the genomic average mutation rate, which is reduced by ~60-fold when the rpoS positive regulator psrA is removed from the genome.

It is surprising that the hotspot mutation is both caused by transcription and causes an increase in transcription. However I think the authors have provided a set of experiments that show that this phenotype is coincidental and that mutation does precede selection. Most reassuring of these results is the Q189W (TGG) mutation, because while this mutation reduces transcription initially, it should not reduce levels of transcription once a reversion to TAG occurs. This strain therefore disentangles mutation rates and phenotype by only affecting transcription prior to the mutational event. This is different from the psrA knockout which removes rpoS transcription both preceding to and following the C565T mutation. My only reservation is therefore not with the effect but with the reported magnitude, as described below:

Major comment (requires clarification or amendment):

Is a fluctuation assay followed by FALCOR analysis an appropriate test for determining the mutation rate of a sequence that is dictated by rpoS induction? RpoS is primarily active/induced in the stationary phase (Schuster et al., 2003; supported by Figure S1), and FALCOR follows the basic assumptions of the Lea‐Coulson Model (Hall et al., 2009). Therefore the assay as employed may violate the following assumptions of Lea-Coulson outlined by Foster, 2006:

1. The cells are growing exponentially. (For this assumption to not be violated the assay must be terminated as soon as exponential phase ends. As the increased mutation rate is driven by the induction of rpoS transcription, and the OD does not change between 18 and 22h (Figure S1), this is not the case.)

2. The probability of mutation per cell-lifetime does not vary during the growth of the culture. (Again, this assumption is violated by mutation rate increasing as rpoS is naturally induced toward the end of the assay).

As the input parameters of FALCOR are the number of mutants (r) within a given cell count (N), if cell divisions continue to occur after the exponential phase but the total N remains unchanged, this may lead to an overestimate in mutation rate. Conversely if mutation rate only increases toward the end of the assay this may lead to an underestimate of mutation rate.

The authors derive a 5700-fold increase based on their fluctuation assay compared against the rate determined by Long et al., 2018. This previous manuscript used a mutation accumulation experiment and determined mutation rate using the number of mutations and the number of elapsed generations. It is therefore not directly comparable to this study for the reasons stated above. I therefore suggest that the reported fold change of 7500 should either be tested more rigorously or clarified.

Could the authors either explain why their 7500-fold report is robust to the violated assumptions outlined above, or add these caveats/limitations when they report the fold change in mutation rate in the manuscript.

Minor comment and typo:

Line 61: Change ‘deamidation’ to ‘deamination’.

Lines 124-125: “a subtle change in the sequence of codon 189 dramatically reduces (but does not abolish) the extreme parallelism previously observed.”

If it is dramatically reduced it can no longer be ‘extreme’. Consider rephrasing to something like: “a subtle change … dramatically reduces the molecular parallelism at the codon, but it continues to mutate at higher frequency than other positions (exact binomial test, p value).”

Reviewer #2: This paper investigates the processes driving the high genetic parallelism observed in cell chain phenotype mutants that evolve in static microcosms of Pseudomonas. The authors suggest that this parallelism is driven by a C-to-T mutational hotspot at position C565 in the nlpD gene, which does not confer a selective advantage. They propose that the elevated C-to-T mutation rate at this site is associated with transcriptional activity of the rpoS gene, whose transcription start site is located within the nlpD gene. The experiments are well done and the writing is lucid. However, I have several concerns about the data and interpretation, noted below.

Major comments

1. I am not convinced that there is no role for selection in explaining the observed genetic parallelism. Fig 2 shows that 5 of the tested nlpD mutations are less fit than C565T and three others have similar fitness. So, none of the tested mutations are more beneficial than C565T. The latter also seems to have a selective advantage compared to the ancestor. Though no statistical test is provided for this comparison, the authors say that the fitness advantage is mild. A mean fitness advantage of ~10%, seen in Fig 2, is far from mild. It is also worth discussing why the results for this competition assay are different from the one shown for the Kan assay (Fig S2, where the ancestor and C565T mutant have similar fitness). Page 9, line 227: the fitness effect of C565T is reported as near-neutral, whereas in Fig S2 the legend says that its fitness is significantly lower. Looking at Fig S2 I am surprised that there is indeed a significantly lower fitness of the C565T mutation; given the error bars I would expect it to be neutral. Please double-check.

Overall, I think all the evidence for a selective advantage of the C565T mutation is a little bit brushed under the rug, and the presented narrative and statements are too strong and not clearly supported by the data. Ideally one would want to try and quantify how much of the parallelism is explained by selection vs. mutation, as done in some earlier studies (though I am not sure this is possible here). As an aside, for the mutants in Fig 2, did these still show the CC phenotype even after the Q189W mutation was reversed?

2. The mutation rate reporter used here is a little problematic. This assay can only detect mutations that will increase rpoS expression (and therefore provide Kan resistance). As a result, any mutations in nlpD that may occur at a comparable rate to C565T, but that have no effect on rpoS expression, will not be detected. So, at the moment it is not clear whether this specific C to T mutation has a higher rate compared to neighbouring positions; and this information is important to attribute the parallelism solely to an elevated mutation rate at that specific site. A second issue is that the mutation rate for C565T from this selection-based mutation rate assay is compared to a genome-wide C to T mutation rate obtained from previous mutation accumulation experiments. Given the large differences in culture conditions and assay methods, I would take the estimated differences in mutation rates with a helping of salt. I don’t think these are lethal problems — I agree that the evidence supports a mutational hotspot in this region — but I think more nuanced discussion, with clarity on the limitations of the assays and results, is warranted.

3. The proposed causal chain of events for high parallelism is not as clear as it could be. Initially I thought that the authors are proposing that because rpoS has generally high transcription, the mutation rate for C565T increases, which increases rpoS transcription further, reinforcing the mutational hotspot. This was puzzling, because the last two steps have no explanatory value for the mutational hotspot idea, since the 565 mutation has already occurred and that’s the end result one is trying to explain. Then I realized that these two things are decoupled, so that from the evolutionary parallelism perspective it is just a coincidence that the C565T mutation also increases rpoS transcription.

If I am understanding this correctly, I suggest that the text make this very clear, so as not to confuse readers.

If I am wrong and the two pieces are in fact connected in the authors’ minds, then those connections should be more clearly laid out, with clear support for the feed-forward loop for the mutational hotspot and its relevance to explain the parallelism.

More generally, if possible it would be good to induce overexpression of rpoS and then test if the mutation rate at C565T increases, as predicted. Providing (e.g., published) data on and discussion of the baseline rpoS transcription level, comparisons with other genes, and changes under experimental conditions, would also be useful.

4. I am a little worried about the variation in the fitness data, which are key to the conclusions. E.g., the very large error bars in the fitness experiments (Fig 2), which in some cases go all the way from s=-0.3 to -0.5 across replicates. Is there a known reason for this large variation across replicates? It appears very high to me, but I could be mistaken, so a comparison with previous studies would be useful. Similarly, the clear marker effects in some cases are a bit worrisome (e.g., Fig S2). Was the latter accounted for in the statistical analysis (i.e., was the marker effect included as a random factor, using a mixed model)?

Minor comments

1. Page 2, line 52: The statement “The mechanistic causes of mutational hotspots are often idiosyncratic and unpredictable” is not well-supported by the following sentences, which discuss common reasons for mutational hotspots. Consider rephrasing to align with the provided examples of not so common mechanisms.

2. Page 3, line 89: The sentence “single codon in nlpD is prone to mutate at high frequency” may be misleading since not all nucleotides in that codon exhibit a high mutation rate. It would be more accurate to say “single nucleotide in nlpD...”

3. Page 5, line 125: The sentence “This shows that a subtle change in the sequence of codon 189 dramatically reduces (but does not abolish) the extreme parallelism previously observed” could suggest that the Q189W mutant may confer some fitness benefit. Could the higher frequency (~20%) of this mutant in the SBW25 ∆wss nlpD Q189W background be due to a slight fitness advantage?

4. Page 12, line 313: Typo: Change “in” to “on”.

5. Line 225: “in the reporter” should be “in nlpD”?

6. This recent paper presents some parallels, and might be relevant to discuss: Rodriguez, A. et al. Synonymous codon substitutions modulate transcription and translation of a divergent upstream gene by modulating antisense RNA production. Proc. Natl. Acad. Sci. 121, e2405510121 (2024).

Reviewer #3: This study by Farr et al. further investigates a frequent mutational site in the nlpD gene of Pseudomonas fluorescens that can be selected by growth in static cultures. This mutation, a C to T transition at nucleotide ?? leads to a “cell chaining” phenotype, due to lack of cell separation at division. Interestingly, this mutation coincides with a predict promoter region for the downstream rpoS gene, encoding the general stress response sigma factor. The authors rule out fitness explanations for the frequent mutation. The data presented here clearly establish that C565T is a bona fide mutational hotspot, whose “hotness” requires transcription. The mechanism of this mutational event will therefore be of general interest. In addition, since many gram negative bacteria carry the same nlpD rpoS arrangement, with a promoter for rpoS internal to nlpD, this could be a more general phenomenon.

specific comments;

line 24 the term“molecular parallelism” is probably not familiar to most readers, please define

line 129 Figure 1: Showing the nucleotide sequence as in Fig S3 would be helpful here.

line 184 please cite stationary phase induction results here, as presented in Fig S1.

line 181 and Fig 3 please explain the relevance of the wss mutation, perhaps briefly in the intro while introducing the Lind et al 2017 results.

line 214 and later. The kan selection results section needs to be expanded. Did all kanR colonies have a mutation in the nlpD region? Results from the Sanger sequencing of kanR isolates should be presented somewhere and the fraction of these with C565T should be stated. A bit more experimental description of how the mutation rate was derived would be helpful in the results section (rather than just in the methods).

line 223 “Interestingly, these fluctuation assays revealed a mutational hotspot slightly broader than base-pair 565 of nlpD” How so? Please show data.

General comments about Discussion and Sankar results.

The authors cite the similarity between this hotspot and one observed in the thyP3 promoter of Bacillus subtilis. The Sankar et al. interpretation, which appears to be shared by the authors of this study, is that transcription promotes mutation, probably by promoting deamination (adenine for Sankar and cytosine here).

However, an alternative explanation (and to my mind, equally likely) is that the hotspot is resistant to correction by cellular surveillance pathways. Indeed the hotspot in the Sankar study goes away in mismatch repair deficient strains, a result consistent with the hypothesis that the hotspot change in not correctable (or corrected incorrectly to the mutated rather than parental base). Especially because there are transcription-coupled repair pathways (nucleotide excision repair, base excision repair and mismatch repair), this should be considered and stated as a possibility. A deaminated cytosine on the non-transcribed strand could be fixed to thymine(uracil) by removal of the parental guanine on the transcribed strand by transcription-coupled mismatch repair. This is why other mutations that arise in the promoter region (see comment above) would be interesting.

Doing the kan selection in a number of interesting mutants (mutL, mfd) would be important. Also , seeing whether C565T is stimulated by nitrous acid (a deamination reagent) and, if so, is it affected by psrA or mutL, may yield some mechanistic insight.

**Have all data underlying the figures and results presented in the manuscript been provided?**

Reviewer #1: Yes

Reviewer #2: Yes

Reviewer #3: None

PLOS authors have the option to publish the peer review history of their article (what does this mean? ). If published, this will include your full peer review and any attached files.

**Do you want your identity to be public for this peer review?** For information about this choice, including consent withdrawal, please see our Privacy Policy .

Reviewer #1: No

Reviewer #2: No

Reviewer #3: No

---

## [Decision Letter · Decision Letter 1]

7 Jan 2025

PGENETICS-D-24-01022R1

An extreme mutational hotspot in nlpD depends on transcriptional induction of rpoS

PLOS Genetics

Dear Dr. Farr,

Thank you for submitting your manuscript to PLOS Genetics. After careful consideration, we feel that it meets our publication criteria. We invite you to submit a revised version of the manuscript that addresses Reviewer #1's minor comments.

Please submit your revised manuscript within 30 days Feb 06 2025 11:59PM. If you will need more time than this to complete your revisions, please reply to this message or contact the journal office at plosgenetics@plos.org. Please include the following items when submitting your revised manuscript:

We look forward to receiving your revised manuscript.

Kind regards,

Ivan Matic

Academic Editor

PLOS Genetics

Lotte Søgaard-Andersen

Section Editor

PLOS Genetics

Aimée Dudley

Editor-in-Chief

PLOS Genetics

Anne Goriely

Editor-in-Chief

PLOS Genetics

**Additional Editor Comments (if provided):**

**Journal Requirements:**

**Reviewers' comments:**

Reviewer's Responses to Questions

**Comments to the Authors:**

Reviewer #1: I thank the authors for addressing the raised comments by myself and the other reviewers. The limitations of the experiments and the remaining questions are now more clearly presented. As for my major comment regarding the reported magnitude of mutation, I support the amendments and additions to the main text and the inclusion of Sup Fig 4. I do have two minor requests, which I believe will increase the clarity of the estimated rate without requiring a litany of additional text:

1. Line 29: "Fluctuation assays demonstrate the C565T mutation rate to be ~5,000-fold higher than expected". Change "demonstrate" to "estimate".

Your reply/sup fig 4 shows that terminating the assay at an earlier timepoint could lower estimated rates by ~7-fold. This is quite significant in the eyes of the reader, as ~5000-fold above average rates falls to ~800-fold. Therefore I think it's important to establish that this fold change is an estimate in the abstract.

2. Line 353: Could a comment be added that links psrA knockout rate to the expected rate, something like: "This represents a 60-fold decrease compared to the rate of mutation in SBW25 (~5000-fold to ~80-fold higher than expected T->C rates)."

The "60-fold reduction" of the psrA knockout is never mentioned in relation to the average rate. I feel the way the rates are currently reported makes them seem high when you want them to and low when you want them to. I.e. the hotspot is ~5000-fold higher than average (makes the rate seem super high) and then psrA knockout is ~60-fold lower than the hotspot (significantly lower, strongly implicating transcription as a causal factor). What is missing is any mention that the psrA knockout is still estimated to be ~80-fold higher than the average rate. Of course this can be readily calculated by the reader but mentioning it seems fair, because at the moment one could incorrectly assume that transcription alone can change mutation rates by ~5000-fold.

Reviewer #2: I am happy with the revisions and I think this paper is a nice contribution. I suggest a thorough reading of the revised text to correct several typos and grammatical errors, but otherwise I have no further comments.

Reviewer #3: My concerns have been addressed.

**Have all data underlying the figures and results presented in the manuscript been provided?**

Reviewer #1: Yes

Reviewer #2: Yes

Reviewer #3: Yes

PLOS authors have the option to publish the peer review history of their article (what does this mean? ). If published, this will include your full peer review and any attached files.

**Do you want your identity to be public for this peer review?** For information about this choice, including consent withdrawal, please see our Privacy Policy .

Reviewer #1: No

Reviewer #2: **Yes: ** Deepa Agashe

Reviewer #3: No

**Figure resubmission:**
---

## [Editor Report · Decision Letter 2]

13 Jan 2025

Dear Dr Farr,

We are pleased to inform you that your manuscript entitled "An extreme mutational hotspot in nlpD depends on transcriptional induction of rpoS" has been editorially accepted for publication in PLOS Genetics. Congratulations!

Yours sincerely,

Ivan Matic

Academic Editor

PLOS Genetics

Lotte Søgaard-Andersen

Section Editor

PLOS Genetics

Aimée Dudley

Editor-in-Chief

PLOS Genetics

Anne Goriely

Editor-in-Chief

PLOS Genetics

Comments from the reviewers (if applicable):

**Data Deposition**

http://datadryad.org/submit?journalID=pgenetics&manu=PGENETICS-D-24-01022R2

**Press Queries**

---

## [Editor Report · Acceptance letter]

PGENETICS-D-24-01022R2

An extreme mutational hotspot in nlpD depends on transcriptional induction of rpoS

Dear Dr Farr,

We are pleased to inform you that your manuscript entitled "An extreme mutational hotspot in nlpD depends on transcriptional induction of rpoS" has been formally accepted for publication in PLOS Genetics! Your manuscript is now with our production department and you will be notified of the publication date in due course.

With kind regards,

Anita Estes

PLOS Genetics

On behalf of:
